# Deploying Reinforcement Learning in Water Transport

**Pak Hei Wong**
Department of Information Engineering
The Chinese University of Hong Kong
Shatin, Hong Kong
`wph019@ie.cuhk.edu.hk`

**Tsz Kui Chow**
Department of Information Engineering
The Chinese University of Hong Kong
Shatin, Hong Kong
`1155093841@link.cuhk.edu.hk`

## Abstract

In this project, we deployed various types of reinforement learning algorithms to resolves the rewards maximization problem in water sailing by reaching the destination with the highest priority using the smallest numebr of steps. We have implemented our own environment for this project and trained different agents using policy iteration, value iteration, and Deep-Q Learning (DQN). We comprehensively evaluates our approach on the environment with $8 \times 8$ and $16 \times 16$ map, our results show that the agents trained by policy iteration and value iteration can reaches the destination by giving certain number of steps, and the agent trained by DQN can finished the sailing using the smaller number of steps.

The implementation code of our project can be accessed using the following link: `https://github.com/donaldphw/IERG5350_project_implementation`.The presentation video of our project can be accessed using the following link: `https://www.youtube.com/watch?v=A0xWcKR70kw`.

## 1 Problem Definition

Route planning in water transport is a challenging task and introduces a significant effect on a shipping company's profit. Given a pre-defined amount of cargoes to a cargo ship located at the starting port, the shipping company needs to design a trajectory to deliver all cargoes to different destinations within a period. Nevertheless, different trajectories may introduce different amounts of cost in time and fuel. Hence, it is crucial to the shipping company to derive an optimal trajectory (the shortest path) which results in the least amount cost to improve productivity, given the list of cargoes and destination pairs.

In this project , we want to investigate the performance of deploying reinforcement learning in optimal trajectory deriving under different conditions such as the number of destinations and the newly added cargoes once the cargo ship arrives at a destination port.

## 2 Related Work

Related works show that reinforcement learning is widely adopted to resolves various problems in logistics. Feng (1) demonstrates that deploying reinforcement learning in the route planning of an unmanned aerial vehicle (UAV) travel. The results show that his approach is able to avoid letting UAV enter multiple no-fly zones.

Kamoshida and Kazama (3) also successfully train an intelligent Automated guided vehicle (AGV) using reinforcement learning techniques to pick items inside a warehouse to reduce the time cost by the picker.

Lu and Han (2) present a model to derive the shortest path to receive and deliver users in an urban environment by considering a high-dimensional logistics environment, to reduce the courier searching time of couriers and waiting time of users to maximize the service output.

## 3    System model

Considering the scenario that total $K$ cargoes are loaded on a cargo ship (agent) at a starting port $S$ and the cargo ship needs to deliver all of them to specific destinations $D_i, i \in K$. In our case, the cargo ship will be filled up with fuel once at the starting port, which allows the cargo to sail in the sea for a limited number of total steps $T$. Considering each cargo has different priority to be delivered under the circumstance at the starting port $S$, each cargo has a different reward $R_i$. The cargo ship can claims the corresponding reward $R_i$ for finished delivering the $i^{th}$ cargo when it reaches $D_i$.

Once the cargo ship reachs any $D_i$, the sailing is finished as the environment needs to be updated as the priority of the undelivered cargoes will change. Also the new cargo may be filled into the cargo ships.

### 3.1    Obstacles

Instead of the starting port $G$, the water $W$, and the destination to be reached $G$, we added an additional state $O$ to denotes the obstacles which a cargo ship can possibly faced during sailing over ship such as islands, lighthouses, or even old rigs. Once the cargo ship crashes on $O$, the sailing terminates immediately.

### 3.2    Total Steps

To motivate the cargo ship to reach any $D_i$ with the shortest path, when it make a move in the sea, the enviroment will introduce $-1$ as reward to the cargo ship. Once the cargo ship has moved $T$ steps over the sea without reaching any $D_i$ or $O$, the sailing terminates as it is out of fuel.

### 3.3    Problem Formulation

Given the above defination, the objective of our works is to maximize the reward $R_i$ the cargo ship can get without reaching any $O$, by giving $D$, $R$, and $T$:

$$max(R_i - t), 0 \leq t \leq T$$

## 4    Environment

To investigate the problem we defined in this project, we need to build an environment to simulate a sea in which the cargo ship will be sailing and keeps giving rewards to the cargo ship based on the travelling steps and the reward $R_i$ which the cargo ship delivered to $D_i$.

We use a 2D grid-map to simluate the enviroment of a cargo ship sailiing in the sea. The movement will be terminated if the cargo crashes on a obstacle grid $O$, reaches any destination $D_i$ , or moves for $T$ steps. Here is the defination of our environment:

- **State**: we use the location (current x position, current y position) of cargo ship over the sea to represent the state
- **Action space**: the directions which a cargo ship is going to ["up", "down", "left", "right"]
- **Maximum steps**: given a pre-defined amout of fuel filled in the cargo ship, it can only sail in the sea for a limited number of steps $T$. We define $T$ as the maximum number of steps which a cargo ship can sail
- **Number of destinations**: there are total $K$ destination port are located in the ground which the cargo ship can reach to deliver the cargoes
- **Reward**: The reward of each states minus the total number of steps has sailed for the cargo ship

**Figure 1: Environment framework for $8 \times 8$ map**

| S -1.0 | W -1.0 | W -1.0 | W -1.0 | W -1.0 | W -1.0 | W -1.0 | D 200.0 |
|---|---|---|---|---|---|---|---|
| W -1.0 | W -1.0 | W -1.0 | O -1.0 | O -1.0 | W -1.0 | W -1.0 | W -1.0 |
| W -1.0 | W -1.0 | W -1.0 | O -1.0 | O -1.0 | W -1.0 | W -1.0 | W -1.0 |
| W -1.0 | W -1.0 | W -1.0 | W -1.0 | W -1.0 | W -1.0 | W -1.0 | W -1.0 |
| W -1.0 | O -1.0 | O -1.0 | W -1.0 | W -1.0 | W -1.0 | W -1.0 | W -1.0 |
| W -1.0 | W -1.0 | W -1.0 | W -1.0 | W -1.0 | O -1.0 | O -1.0 | W -1.0 |
| W -1.0 | W -1.0 | W -1.0 | W -1.0 | W -1.0 | O -1.0 | O -1.0 | W -1.0 |
| D 400.0 | W -1.0 | W -1.0 | W -1.0 | W -1.0 | W -1.0 | W -1.0 | D 1000.0 |

**Figure 2: Environment framework for $16 \times 16$ map**

| S -1.0 | W -1.0 | W -1.0 | W -1.0 | W -1.0 | W -1.0 | W -1.0 | W -1.0 | W -1.0 | W -1.0 | W -1.0 | W -1.0 | W -1.0 | W -1.0 | W -1.0 | D 4000.0 |
|---|---|---|---|---|---|---|---|---|---|---|---|---|---|---|---|
| W -1.0 | W -1.0 | W -1.0 | W -1.0 | W -1.0 | W -1.0 | W -1.0 | W -1.0 | W -1.0 | W -1.0 | W -1.0 | W -1.0 | W -1.0 | W -1.0 | W -1.0 | W -1.0 |
| W -1.0 | W -1.0 | W -1.0 | W -1.0 | W -1.0 | W -1.0 | W -1.0 | W -1.0 | W -1.0 | W -1.0 | W -1.0 | O -1.0 | O 0.0 | W -1.0 | W -1.0 | W -1.0 |
| W -1.0 | W -1.0 | W -1.0 | W -1.0 | O -1.0 | O -1.0 | O 0.0 | W -1.0 | W -1.0 | W -1.0 | W -1.0 | O -1.0 | O 0.0 | W -1.0 | W -1.0 | W -1.0 |
| W -1.0 | W -1.0 | W -1.0 | W -1.0 | O -1.0 | O -1.0 | O 0.0 | W -1.0 | W -1.0 | W -1.0 | W -1.0 | W -1.0 | W -1.0 | W -1.0 | W -1.0 | W -1.0 |
| W -1.0 | W -1.0 | W -1.0 | W -1.0 | W -1.0 | W -1.0 | W -1.0 | W -1.0 | W -1.0 | W -1.0 | W -1.0 | W -1.0 | W -1.0 | W -1.0 | W -1.0 | W -1.0 |
| W -1.0 | O -1.0 | O 0.0 | W -1.0 | W -1.0 | W -1.0 | W -1.0 | W -1.0 | W -1.0 | W -1.0 | W -1.0 | W -1.0 | W -1.0 | O -1.0 | O -1.0 | O -1.0 |
| W -1.0 | O -1.0 | O 0.0 | W -1.0 | W -1.0 | W -1.0 | W -1.0 | W -1.0 | W -1.0 | W -1.0 | W -1.0 | W -1.0 | W -1.0 | O -1.0 | O -1.0 | O -1.0 |
| W -1.0 | W -1.0 | W -1.0 | W -1.0 | W -1.0 | W -1.0 | W -1.0 | W -1.0 | W -1.0 | W -1.0 | W -1.0 | W -1.0 | W -1.0 | W -1.0 | W -1.0 | W -1.0 |
| W -1.0 | W -1.0 | W -1.0 | W -1.0 | W -1.0 | W -1.0 | O -1.0 | O -1.0 | O -1.0 | W -1.0 | W -1.0 | W -1.0 | W -1.0 | W -1.0 | W -1.0 | W -1.0 |
| W -1.0 | W -1.0 | W -1.0 | W -1.0 | W -1.0 | W -1.0 | O -1.0 | O -1.0 | O -1.0 | W -1.0 | W -1.0 | W -1.0 | W -1.0 | W -1.0 | W -1.0 | W -1.0 |
| W -1.0 | W -1.0 | W -1.0 | W -1.0 | W -1.0 | W -1.0 | W -1.0 | W -1.0 | W -1.0 | W -1.0 | W -1.0 | W -1.0 | W -1.0 | W -1.0 | W -1.0 | W -1.0 |
| W -1.0 | O -1.0 | O -1.0 | W -1.0 | W -1.0 | W -1.0 | W -1.0 | W -1.0 | W -1.0 | W -1.0 | W -1.0 | W -1.0 | W -1.0 | W -1.0 | W -1.0 | W -1.0 |
| W -1.0 | O -1.0 | O -1.0 | W -1.0 | W -1.0 | W -1.0 | W -1.0 | W -1.0 | W -1.0 | W -1.0 | O -1.0 | O -1.0 | O -1.0 | W -1.0 | W -1.0 | W -1.0 |
| W -1.0 | W -1.0 | W -1.0 | W -1.0 | W -1.0 | W -1.0 | W -1.0 | W -1.0 | W -1.0 | W -1.0 | O -1.0 | O -1.0 | O -1.0 | W -1.0 | W -1.0 | W -1.0 |
| D 8000.0 | W -1.0 | W -1.0 | W -1.0 | W -1.0 | W -1.0 | W -1.0 | W -1.0 | W -1.0 | W -1.0 | W -1.0 | W -1.0 | W -1.0 | W -1.0 | W -1.0 | D 20000.0 |

Figure 1 and figure 2 illustrates the map which the cargo ship is going sail in our environment. We starts with a $8 \times 8$ map as our baseline and later extend the scale to a $16 \times 16$ map. Both maps have $K = 3$ destinations. To give different priorities to the cargo ship to reach different $D_i, i \in K$, the rewards $R_i$ for each $D_i$ are different. We have implemented two environment: **SailingEnv** forr policy iteration and value iteration, and **SailingEnvDQN** for DQN.

## 5 Methodology

### 5.1 Value Iteration

Since the location of $S$, $O$ and $K$ are fixed by the cargo ship before it starts sailing, and we need to derive the shortest path to reach a $D_i$ which has the highest reward $R_i$ and we assume such path exists. Hence, there must be a optimal action to take at each stage to achieve the shortest path. Given the maximum steps to take $T$, the location and reward $R_i$ for each $D_i$, we can convert our task into an optimizaton problem: maximizing $R_i$ and minimizing $t$.

Since, such optimal value function exists in our assumption, we can deploy value iteration to solve our problem.

### 5.2 Policy Iteraction

Since we assume there exists an optimal policy which has a greater value than the previous one, we can also deploy policy iteration to solve our problem.

### 5.3 Deep Q-Learning

The DQN we built for this project is constructed by one hidden layer which has 128 neurons, followed by a ReLU layer. We deployed Adam optimizer, MSE loss function, and clip gradient during training.

## 6 Experiments

We comprehensively evaluated our agents trained by different RL algorithms in both $8 \times 8$ and $16 \times 16$ maps, the records, then we recorded the mean rewards and mean number of steps for each agents over both maps. The ultimate goal for the agent is to find out the smallest number of steps to reach $D_i, i \in K$, which has the highest $R_i$. In order to experiement on the effect of different $T$, we evaluate our agents by setting different $T$ and see if they introduce better results or not in different values.

### 6.1 Mean reward

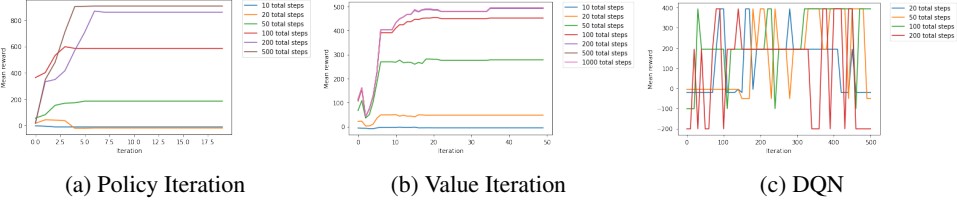

(a) Policy Iteration         (b) Value Iteration         (c) DQN

Figure 3: Mean reward of training ($8 \times 8$ map)

Figure 3 illustrates the mean rewards introduced by agents running in a $8 \times 8$ map trained by different RL algorithms, by giving different values of $T$. The results show that a larger $T$ gives positive effect to the agents to explore higher mean rewards for the agent trained by policy iteration and value iteration. On the other hand, the agents trained by DQN trends to finding paths whether for running all $T$ steps or reaching any $D_i$ between the one in top right corner and the one in the bottom left corner, and not converging to a point. Overall, the agents trained policy iteration achieve the highest mean rewards in the $8 \times 8$ map.

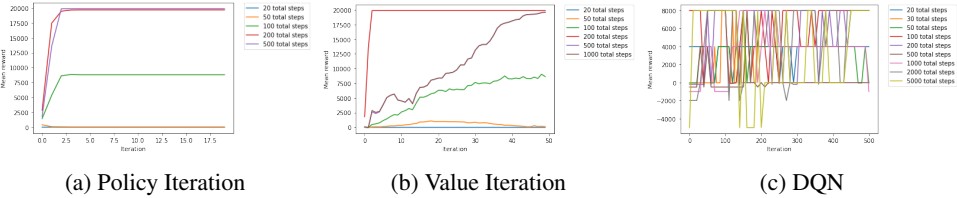

(a) Policy Iteration       (b) Value Iteration       (c) DQN

Figure 4: Mean reward of training ($16 \times 16$ map)

Similar results to $8 \times 8$ map can be observed in $16 \times 16$ map in Figure 4 . However, a difference is that by given enough $T$ to the agents trained by value iteration, they can achieve mean rewards as high as the mean rewards achieved by the agents trained by policy iteration.

## 6.2 Mean Step

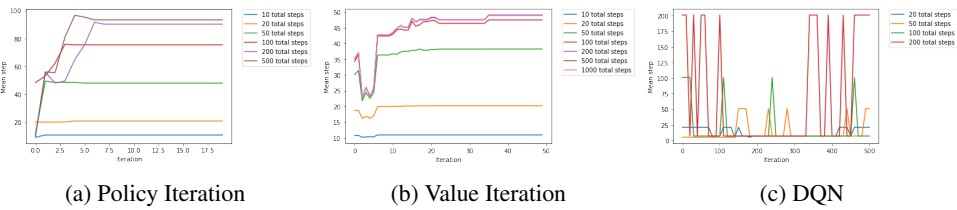

(a) Policy Iteration       (b) Value Iteration       (c) DQN

Figure 5: Mean step of training ($8 \times 8$ map)

Figure 5 demonstrates the mean step achieved by agents trained by different RL algorithms by giving different $T$ in the $8 \times 8$ map. Comparing the results in section 6.1, obtaining a higher mean reward requires more number of steps for the agents trained by policy iteration and value iteration. The results also show that the agents trained by value iteration require slightly less steps to achieve the optimal mean reward, compare with the results of the agents trained by policy iteration.

On the other hand, the agents trained by DQN shows dissimilar but interesting results compare with the other two. The results of DQN are not converging to a optimal points, but randomly result in crashing on a obstacle $O$ using the path (using the smallest steps to finish the episode), reaching the $D$ located in the top right corner and bottom left corner, or running all $T$ steps without reaching any $O$ or $D_i$.

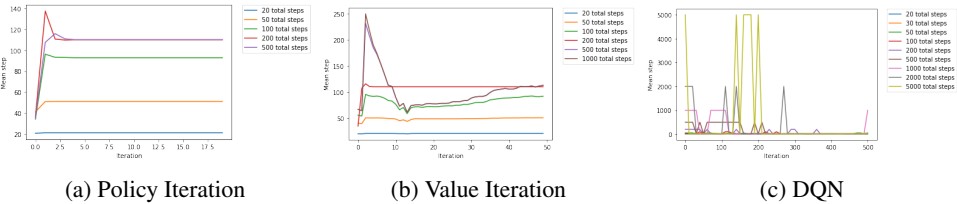

(a) Policy Iteration       (b) Value Iteration       (c) DQN

Figure 6: Mean step of training ($16 \times 16$ map)

Figure 4 shows that the mean steps resulted by agents running in $16 \times 16$ map is similar to the results in $8 \times 8$ map. The only difference is that the mean steps in $16 \times 16$ rise to a peak at the beginning during training, then fall back to a optimal value and converge, in the setting with policy iteration and value iteration.

# 7 Conclusion

In this project, we develop our own environment to simulate the environment of a cargo ships which needs to deliver cargoes with different priority using the smallest number of steps without crashing on any obstacle over the sea. Then we train agents using policy iteration, value iteration, and DQN to explore in this environment to see if they can derive the shortest path to reach the destination which has the highest reward. The results in policy iteration and value iteration shows that the agents trained by the above RL algorithms is capable to derive the path to reach the destination which has the highest reward, but fail to procure the shortest path. The agents trained by DQN is able to find a shortest path to finish the episode, or reaching the destination located in the top right and bottom left corner, or moving for $T$ steps without crashing on any obstactle or reaching any destination.

# 8 Future work

At the beginning, we thought DQN should be able to overkill this problem, but the result is not. One possible reason may be the rewarding mechanism in SailingEnvDQN, which we are going adjust. Also, we are going to experiment on a more complex DQN as the one we used in this project is quite simple.

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
