# OpenReview forum: "Deploying Reinforcement Learning in Water Transport"
_CUHK.edu.hk/2021/Course/IERG5350_

### Official Review · AnonReviewer1 · 2020-12-16
**A decent extension to the FrozenLake-v0 environment for water transport**

**Rating:** 6
**Confidence:** 5

**Review:**

__General__

The project simulates a worldwide water transport problem with reinforcement learning with an environment extending FrozenLake-v0. To obtain an optimal route, the authors implemented vanilla policy iteration, value iteration, and DQN algorithms. All 3 algorithms are evaluated with different combinations of hyperparameters. The policy iteration method can achieve a sound reward in the given environment and the potential reasons behind the performance of 3 algorithms are analyzed.

__Recommendation__

The project defines the problem clearly and presents an acceptable result. However, current environment simulation is not very applicable to the real world because the obstacles and destinations are determined by the authors instead of simulating the real world map. The same problem occurs in scalability which the authors said will improve in the future. It will be helpful to revisit the talk by Tony Qin from DiDi Research as he mentioned a similar problem.

---

### Official Review · AnonReviewer2 · 2020-12-16
**Deploying Reinforcement Learning in Water Transport**

**Rating:** 8
**Confidence:** 5

**Review:**

summary: This paper deploys different kinds of reinforcement learning algorithms to solve the water sailing problem.

Clarity: Clear problem definition and environment description.
pos:
1. open-source implementation code.
2. provide an intuitive environment framework.
3. use different kinds of methods to solve the problem.
4. give two metrics to evaluate the performance
Cons:
1. no detailed configurations about value iteration and policy iteration are provided.
2. more analyses about the comparisons.

---

### Official Review · AnonReviewer3 · 2020-12-20
**Comparison between Policy Iteration, Value Iteration and DQN on classical RL environment Gridworld**

**Rating:** 3
**Confidence:** 5

**Review:**

**General:**

This paper compares the performance of policy iteration, value iteration and DQN on a classical RL environment Gridworld. However, the authors doesn't show the evaluation results in experiments.

**Quality:**

Only 3 related works are listed, which is not enough. And in experiments, it seems that their conclusion is **based on the performance in training phase**, as they doesn't put any evaluation result. However, as we know for methods like DQN which adpots epsilon-greedy, they introduce randomness in training phase. If the authors forgot to put the evaluation results, please add it in the revision phase, and I'll consider adjusting my rating.

**Clarity:**

The paper is easy to follow.

**Originality:**

Neither environment nor methods are novel. Though the authors describe it as Water Transport problem, it's actually the classical Gridworld problem.

**Significance:**

No significance, unless the authors provide more experiments to prove their conclusion is correct in the revision phase.

**Pros:**
1. They design a certain Gridworld environment and compare the performance of different RL algorithms.
2. They compare both steps and mean reward for different RL algorithms training in different size of environment.

**Cons:**
1. Please extend the related work, where only 3 works listed currently.
2. There is no evaluation result in the experiments.
3. There are many typos, please find in the comments.
4. In P5. Figure 4. (b) The red curve is quite interesting. But the authors doesn't mention why it converges so fast.
5. The authors conclude that the agents trained by DQN can find the shortest path while the agents trained by policy iteration and value iteration can find the path maximizing the reward. However there is no evidence to prove that. Since from the training curve, we can't know whether the agent crashes the obstacles, leading to smaller steps.

**Comments:**
1. In P2. Section 3. "The cargo ship can claims" should be "The cargo ship can claim".
2. In P2. Section 3.3. "defination" should be "definition".
3. In P2. Section 4. "sailiing" should be "sailing".
4. In P2. Section 4. "defination" should be "definition".
5. In P3. Figure 1. Figure 2. The size is too large. Please scale it to 50% of the current size.
6. In P4. Section 4. "starts" should be "start".
7. In P4. Section 4. "forr" should be "for".
8. In P5. Section 6.2. "compare" should be "comparing".